# Is It Possible to Predict COVID-19? Stochastic System Dynamic Model of Infection Spread in Kazakhstan

**DOI:** 10.3390/healthcare11050752

**Published:** 2023-03-03

**Authors:** Berik Koichubekov, Aliya Takuadina, Ilya Korshukov, Anar Turmukhambetova, Marina Sorokina

**Affiliations:** 1Department of Informatics and Biostatistics, Karaganda Medical University, Gogol St. 40, Karaganda 100008, Kazakhstan; 2Institute of Life Sciences, Karaganda Medical University, Gogol St. 40, Karaganda 100008, Kazakhstan

**Keywords:** COVID-19, system dynamic, SIR model, forecasting

## Abstract

Background: Since the start of the COVID-19 pandemic, scientists have begun to actively use models to determine the epidemiological characteristics of the pathogen. The transmission rate, recovery rate and loss of immunity to the COVID-19 virus change over time and depend on many factors, such as the seasonality of pneumonia, mobility, testing frequency, the use of masks, the weather, social behavior, stress, public health measures, etc. Therefore, the aim of our study was to predict COVID-19 using a stochastic model based on the system dynamics approach. Method: We developed a modified SIR model in AnyLogic software. The key stochastic component of the model is the transmission rate, which we consider as an implementation of Gaussian random walks with unknown variance, which was learned from real data. Results: The real data of total cases turned out to be outside the predicted minimum–maximum interval. The minimum predicted values of total cases were closest to the real data. Thus, the stochastic model we propose gives satisfactory results for predicting COVID-19 from 25 to 100 days. The information we currently have about this infection does not allow us to make predictions with high accuracy in the medium and long term. Conclusions: In our opinion, the problem of the long-term forecasting of COVID-19 is associated with the absence of any educated guess regarding the dynamics of *β(t)* in the future. The proposed model requires improvement with the elimination of limitations and the inclusion of more stochastic parameters.

## 1. Introduction

Since the start of the COVID-19 pandemic, scientists have to actively used models to determine the epidemiological characteristics of the pathogen, the speed of its spread and its impact on the healthcare system and state policy in the medical field. In a literature review [1], the authors note that 920 research articles that had been published by 10 October 2020 were found concerning the problem of predicting COVID-19. Another review is based on 1196 publications posted in various databases (Google Scholar, Web of Science and Scopus) from January to December 2020 [2]. An assessment of the reliability and accuracy of models based on different approaches can be found in a systematic review involving 242 articles from 1 January 2020 to 30 November 2020 [3]. A similar review of the models applied in various European countries is given in [4]. Aniruddha, A., et al. analyzed some of the mathematical models implemented to support pandemic response efforts in the US and Sweden [5]. The experience of applying mathematical modeling to the pandemic in India is presented in the publication of Abhinav Gola et al. [6].

The authors of all the above articles are unanimous in their opinion that all known mathematical and computer approaches have been used to model the spread of COVID-19. Among them are statistical analysis, regression analysis, epidemiological SIR and SEIR models and their modifications, agent-based modeling, system dynamics, machine learning methods, etc. Most studies were carried out in Asian (mainly in China and India) and European countries. Most researchers used compartmental models (SIR and SEIR) and statistical models (exponential growth models and time series), while a few used artificial intelligence, periodogram-based analysis, the Bayesian approach, network models and agent-based models [7,8].

The goals of creating models can be very different; some focus on risk factors for death from COVID-19 or comorbidities and others on the effectiveness of diagnostic tests. A significant amount of work is focused on considering various scenarios for the development of a pandemic in the context of lockdown, vaccination and other preventive measures. Thus, in the review [3], out of 555 models included in the paper, only 254 were devoted to predicting morbidity. The authors discuss the question of how the forecasts obtained with the help of these models come true and how large the differences between the forecast and real data are. The quality of forecasts can be affected by the type of model, the completeness and availability of epidemiological data, the level of quarantine measures and other factors. It can be seen there that, out of 59 predictions, in 38% of cases, the predicted values were higher than the real ones, and in 62% of cases, they were lower than the observed values. No differences in accuracy were found between different categories of models nor within each category.

However, Shakeel, S.M., et al. [2] point to the high accuracy of forecasting models based on machine learning. Supporting this view, in [9], it was shown that the decision tree model had the highest accuracy of 94.99%, while the support vector machine model had the highest sensitivity of 93.34% and the naïve Bayes model had the highest specificity of 94.30%. Sufficient accuracy for practical use was obtained in the K-nearest neighbors and gradient boosting models [10] and in the regression model [11]. The model using the XGBoost algorithm had an accuracy of 90% [12,13], and the random forest method made it possible to predict confirmed cases with high accuracy [14,15].

In another review, the author analyzed the accuracy of 51 US CDC COVID-19 forecasting models [16]. The models included an analysis of four pandemic waves from 6 July 2020 to 17 January 2022. The average forecast error during the first wave for SEIR models was 31%, and for machine learning models, it was 32%. During the second wave, it increased to 45% and 44%, then to 63% and 63% and, during the fourth wave, to 92% and 44%, respectively. Thus, the prediction error increased with increasing prediction time. The authors attribute this to changes in real-world conditions, such as the emergence of new variants, the development or loss of immunity, higher R0, new restrictive measures and the development of vaccines.

One publication focused on the accuracy of COVID-19 prediction models in Sweden. The authors concluded that these models had limited accuracy—only 4 out of 16 models had a prediction error of less than 20%. It was assumed that the main reasons for this are the lack of reliable data in the initial period of the pandemic, as well as the fact that the works were published before the forecast period, which did not allow for the comparison of model data with real values [17].

The SIR model and its modifications also suffer from insufficient forecasting accuracy. The article [18] emphasizes that the failure of SIR-based models to predict the COVID-19 pandemic could have been caused by a number of reasons. First, these overly simplistic models ignore factors that have a significant impact on the course of the disease. Another explanation for the inability of SIR-based models to predict COVID-19 epidemics is that the modeling is based on assumptions that are not necessarily correct.

The article [19] discusses three models created in the UK, US and Austria and used in the formation of public policy in the fight against the pandemic. It is noted that all models are useful in terms of creating and analyzing various “what if” situations. They are a way of informing the public and the government about the extent of the problems associated with the pandemic. At the same time, the authors believe that these models cannot be considered as a tool for long-term forecasting (for months ahead), since there are no credible estimates of their validity. The problem of long-term forecasting is also pointed out in [13]. Although the model considered in the publication made it possible to predict the nature of the course of the disease in the short term, the predicted values were far from real data in later time intervals, and the constructed model did not allow prediction of the course of the epidemic in the long term.

Relatively satisfactory results were obtained with short-term forecasting. For several countries, the SIR models showed a forecast of total cases with an error of 5%. However, the authors of the study note that the results were reliable only in cases where the spread of the epidemic had ended, i.e., at the maximum point or at the inflection point. Verification of accuracy by comparing forecasts with actual data showed that the forecasts were realistic as long as lockdown measures were maintained [20]; a similar result is given in [21,22]. Since there was an inflection point on the total cases curve, when using the SIRD model, the predicted data deviated significantly from the real data.

It is worth noting that the researchers used various predictors to forecast outbreaks of COVID-19 infection. Among them were total cases, mortality rate [23] and reported daily cases [24,25,26]. New pandemic descriptors were also developed to predict outbreaks, e.g., transmission index [27] or infection rate [28]. However, this did not improve the accuracy of forecasting [3].

Kazakh scientists have published a number of works on the modeling of the COVID-19 epidemic. At the same time, the researchers pursued various goals: the assessment of risk factors [29,30], forecasting the needs of the healthcare system in material and labor resources [31] and forecasting morbidity [32]. The forecast made in [33] turned out to be far from reality: based on simulated data, it was projected that within six months the number of infected individuals in Kazakhstan would be 982,010, but, in reality, it was 11,239.

Most of the models mentioned above were created in the early months of the pandemic. When predicting, the researchers proceeded from the assumption of a further increase in the number of new cases. Associated with this was a discrepancy between expected and observed rates of morbidity, mortality and “recovery”. The development of events during 2020 and 2021 has shown that the dynamics of daily incidence is oscillatory, and periods of relative stability are accompanied by sudden outbreaks, the nature of which has not yet been explained. Transmission rate, recovery rate and the loss of immunity to the COVID-19 virus change over time and depend on many factors, such as the seasonality of pneumonia, mobility, testing frequency, the use of masks, the weather, social behavior, stress, public health measures, vaccination, etc. These rates are difficult to estimate and cannot be estimated from laboratory experiments or surveys of infected populations.

In this regard, a number of authors consider the stochastic models of the spread of COVID-19. Among them are the stochastic matrix, which describes a stochastic process known as a Markov process [27], and stochastic calculus [34,35], which includes differential equations and integrals depending on stochastic processes, such as the Wiener process, also called the Brownian motion (white noise) process [36,37,38]. Hoertel et al. developed a stochastic agent-based model of various scenarios of preventive measures [39]. It must be taken into consideration that the process variability can be especially important when populations are small or certain events occur very rarely [40]. Stochastic models include several factors and are therefore more realistic. Thus, using a stochastic mathematical model, the influence of the environment on the spread of COVID-19 was studied in [41]. A stochastic epidemiological SEIR model with random fluctuations of parameters was used to analyze the dynamics of COVID-19 in Bogota (Colombia) [42]. In [43], a discrete-time stochastic epidemic model with binomial distributions was developed to study the transmission of the disease in China. In one of the publications, the stochastic model was created by expanding the deterministic model by introducing the intensity of stochastic factors and Brownian motion. In that paper, the authors conducted a study to develop a stochastic SVITR mathematical model for the transmission dynamics of the COVID-19 pandemic by introducing treated and vaccinated classes [44,45]. A similar approach was used in the analysis of a stochastic model of COIVD-19 in the Greater Abidjan Region. White noise and jumps were introduced, which corresponded to the different disturbances that can occur [46].

Another study concerns the simulation of the spread of the COVID-19 disease in a community by applying the Monte Carlo method to a susceptible–exposed–infective–recovered (SEIR) stochastic epidemic model [47]. Stochastic models were developed in which the intensity of contact displayed a Poisson distribution and the contact time displayed an exponential distribution [48]. In one of the publications, a stochastic model for the relaxation of lockdown that considered the re-opening of society in terms of people moving between sites on a one-dimensional grid was performed [49].

The stochastic model displays multiple properties of COVID-19. Among these are the high uncertainty regarding the evolution of the outbreaks, long periods in which the disease runs undetected, sudden disappearances followed by outbreak and the unimodal and bimodal progressions of daily disease cases [50].

It is known that the quality of any model depends on the quality of the real data that underlies it and on the accuracy of the selection of the parameters of the model. Many published models were based on data from 2020, when we did not yet have enough information regarding the patterns of the spread of COVID-19. Perhaps this is one of the reasons for the failure of the long-term forecasting of the pandemic. To date, we have already accumulated significant material on the dynamics of this infection, and it is obvious that it proceeds differently in different countries and depends on many factors, the intensity of which is difficult, if not impossible, to measure. Many of the scenarios presented in the literature are hypothetical and do not always correlate with the real situation. Therefore, in our opinion, stochastic models are more realistic, as they also allow the creation of optimistic, pessimistic and most probable scenarios.

We have data on daily cases and hospitalizations for the cities of Kazakhstan for 2020 and 2021. In this period, there were both periods of relative stability and waves of morbidity.

First, we set ourselves the goal of estimating model parameters from this large database. In this case, the key parameter was the transmission rate. 

Secondly, the task was to establish whether this would improve the accuracy of forecasting in the short and long term. 

Thirdly, this work was carried out within the framework of a grant allocated by the Ministry of Health of the Republic of Kazakhstan, which is aware that the availability of reliable forecasts is an important factor in making timely regulatory decisions. We are, therefore, looking for opportunities to create forecasting models that consider the specifics of the socio-economic situation in Kazakhstan, which could be the basis for other models in the event of outbreaks of new infectious diseases similar to COVID-19.

Many studies of the stochastic processes of infection spread are based on epidemiological SIR and SEIR models. Some authors believe that the SEIR model is more suitable for forecasting [51,52], but there is an opinion that more complex models may not necessarily be more reliable in making predictions due to the larger number of model parameters to be estimated [53]. In this paper, we consider SIR models that are focused on counting the number of individuals in each of the states (compartments). We created a model of the spread of COVID-19 based on system dynamics. The key stochastic component of our modified SIR model is the transmission rate, which we consider as an implementation of Gaussian random walks with unknown variance, which we must learn from real data.

## 2. Materials and Methods

### 2.1. Developing a System Dynamic Model

To achieve the objectives of the study, a model of infection spread in a large settlement of Kazakhstan—Karaganda city, with a population of 500,896—was created. We used a system dynamic approach with the SIR technique (susceptible–infectious–recovered). The simplest compartmental SIR model was proposed by Kermack and McKendrick in 1927 [54], which we modified and supplemented with symptomatic flows moving toward outpatient and inpatient treatment (Figure 1), i.e., symptomatic individuals are immediately isolated and cannot infect the susceptible.

According to the SIR model, the population is divided into three distinct classes: the susceptible (*S*), healthy individuals who are able to catch the disease; the infectious (*I*), namely those who have the disease and can transmit it; and the recovered (*R*), i.e., individuals who have had the disease and are now immune to the infection (or removed from further propagation of the disease by some other means).

The equations of this model are as follows: $$\frac{dS}{dt}=-\frac{\beta I\left(t\right)S\left(t\right)}{N}$$
$$\frac{dI}{dt}=\frac{\beta I\left(t\right)S\left(t\right)}{N}-\gamma I\left(t\right)$$
$$\frac{dR}{dt}=\gamma I\left(t\right)$$
where *S(t)* is the number of susceptible cases, *I(t)* is the number of infected cases and *R(t)* is the number of recovered cases and deaths, respectively, at time *t*. *N* is the population size, *𝛽* is the transmission rate and *𝛾* is the probability of the infected cases withdrawing from the epidemiological system (mortality rate plus cure rate).

From the point where symptoms of COVID-19 appeared (after the incubation period), patients were isolated in families or hospitalized and were no longer a source of infection. In our model, we designated *R(t)* as the number of removed cases and *𝛾* as the removal rate.

Model equations (Figure 1):$$\frac{d\left(Susceptible\right)}{dt}=-InfectedRate$$
InfectedRate=Beta∗Infectious∗Susceptible/Population
$$\frac{d\left(Infectious\right)}{dt}=InfectedRate-SymptomsRate$$
$$SymptomsRate=delay\left(InfectedRate,\;IncubTime\right)$$
$$\frac{d\left(Symptoms\right)}{dt}=\left(SymptomsRate-HospitalRate-AmbulRate\right)$$
AmbulRate=SymptomsRate∗0.73
HospitalRate=SymptomsRate∗0.27
$$\frac{d\left(Ambulatory\right)}{dt}=AmbulRate$$
$$\frac{d\left(Hospital\right)}{dt}=HospitalRate$$

According to the Ministry of Health of the Republic of Kazakhstan, on average, 73% of patients with COVID-19 were on outpatient treatment and 27% were hospitalized.

Certain limitations were imposed on the model:–The recovered population do not become susceptible again.–The symptomatic are immediately isolated and cannot infect the susceptible.–Population is not affected by birth and death rates.–The incubation period for all infected individuals is 6 days.

In the literature, data on the incubation period vary widely: from 1.8 to 9.0 days according to a review [55] or from 1 to 14 days according to [56]; the average value was 5.8 days (95% CI from 5.0 to 6.7) in [57]. In review [58], 53 articles were included in the meta-analysis. The pooled mean incubation period of COVID-19 was 6.0 days (95% CI 5.6–6.5) globally. We set an incubation period of 6 days. In the more-recent literature (for 2022), updated data on the incubation period for various unique strains and various age groups has been published [59,60]. We discuss this issue in Section 4.2.

The model was implemented in the AnyLogic University Researcher 8.8.1 software. Chicago, IL, USA.

### 2.2. Data Collection

Data on the daily and total cases of COVID-19 between 10 March 2020 and 31 December 2021 (660 days) were collected from the information system of the Ministry of Health of the Republic of Kazakhstan (Figure 2a,b).

### 2.3. The Estimation and Prediction Scheme

We used three training time frames to define certain statistical characteristics:–The first training time frame (AC) included periods of relative stability in daily new cases, as well as periods of rise and fall in incidence. This corresponds to the first 300 days of the pandemic development in Karaganda from 10 March 2020 to 5 January 2021.–The next training time frame (AD) was 400 days from the first recorded COVID-19 incident in the city (from 10 March 2020 to 15 April 2021, see Figure 2a). This period includes the first wave and the beginning of the second wave from about 361 days.–The third training time frame (AF) was 500 days, from 10 March 2020 to 24 July 2021. In this period, there were two waves of morbidity and the rise of the third—the most “powerful” wave.

Based on each of these time periods, total cases were predicted for the next 100 days according to the following steps:

Step 1. We calculated daily transmission rates in the training time frame using daily data on the number of infectious and susceptible individuals:$$\beta_{t}=-\frac{N\left(S\left(t\right)-S\left(t-1\right)\right)}{S\left(t-1\right)I\left(t-1\right)}$$

Step 2. The distribution function for the transmission rate was estimated, and the mean value and standard deviation in each training time frame were calculated.

Step 3. We used the SD model to make predictions for 100 days. We accepted the transmission rate *β(t)* as changing in time and realizing a random process that reflects changes in the behavior of the population over time.

Step 4. To see the effect of the stochastic variation of the transmission rate, Monte Carlo simulation methods were carried out, with each run of the model containing 100 iterations. In each iteration, a different value from within the defined range for the transmission rate was used. This produced output values that are value ranges rather than point estimates. Next, the minimum, maximum and mode values (optimistic, pessimistic and most likely scenarios) for each run were found.

Step 5. We used the mean absolute percentage error (MAPE) metric for contrasting between predicted and true values. MAPE was chosen because of its scale independence property, which would remove bias due to the size of the test data from the models. MAPE is a measurement of how accurate a forecasting model is and gives its output as a percentage value. It is calculated by normalizing the average error at each point.
$$\mathrm{MAPE}=\frac{1}{n}\overset{n}{\underset{t=1}{\sum }}\left|\frac{A_{t}-F_{t}}{A_{t}}\right|$$
where *A_t_* is the actual data, *F_t_* is the forecast data at time *t* and *n* is the number of forecast days.

To assess the accuracy of daily forecasts, the absolute percentage error (APE) was calculated:$$\mathrm{APE}=\left|\frac{A_{t}-F_{t}}{A_{t}}\right|$$

Model accuracy was evaluated according to the criteria in [60] as highly accurate (MAPE% < 10), good (MAPE%: 10–20), reasonable (MAPE%: 20–50), and inaccurate forecasting (MAPE% > 50).

## 3. Results

There was a relatively low incidence of COVID-19 (incidence rate of 0.5240 per 1000 population) in Kazakhstan due to the tough preventive measures taken by the government at the beginning of the pandemic. Figure 2a shows that, in Karaganda, periods of relative stability were followed by sharp surges in incidence. We can see at least three peaks in this diagram. The amplitude of these peaks is different, and the time intervals between them also involve different durations. All this indicates that the transmission rate *β* changed over time, either increasing or decreasing, despite the high level of stringency of preventive measures (Figure 3). 

According to our calculations, in the first transmission rate training time frame, *β(t)* was normal distributed with a mean value of 0.170 and a standard deviation of 0.075 (Figure 4).

Similarly, these parameters were calculated in the second and third training time frames. The results are presented in Table 1.

Based on these results, we simulated the infection spread over the next 100 days using the stochastic value of *β*:

In the first testing time frame (from day 301 to day 400),
$$\beta \left(t\right)=0.17+N\left(0.075,\;0\right)$$

In the second testing time frame (from day 401 to day 500),
$$\beta \left(t\right)=0.177+N\left(0.074,\;0\right)$$

In the third testing time frame (from day 501 to day 600),
$$\beta \left(t\right)=0.182+N\left(0.073,\;0\right)$$

In the first test time frame (from days 300 to 400), total cases were predicted using the Monte Carlo procedure, and three scenarios were identified: an optimistic scenario with a minimum number of total cases (green line, Figure 5a), a pessimistic scenario with a maximum number of total cases (orange line) and the most probable scenario (red line). The predicted data were compared with the actual data (blue line). Forecasting accuracy (APE) for each of these days for all three scenarios is shown in Figure 5b. Several important conclusions can be drawn from these plots: 

Real data turned out to be outside the predicted minimum–maximum interval. The minimum predicted values of total cases were closest to the real data. 

In the first testing time frame, the forecast APE of the optimistic scenario was less than 10% over all 100 days. In the pessimistic scenario, APE increased from 3% to 67%, and in the most probable scenario, it increased from 3% to 31% within 100 days of forecasting.

To uncover how the MAPE (%) changes with an increase in the duration of the forecast, we divided the time interval of 100 days into four sections, and then, the forecasting periods were 25, 50, 75 and 100 days (Table 2a). Highly accurate forecasting was obtained under the optimistic scenario; highly accurate and good forecasting under the most likely scenario; and high, good and satisfactory results under the pessimistic scenario.

In the second test time frame (from 400 to 500 days), a similar simulation was carried out, the results of which are shown in Figure 5c,d. It can be seen that, for all scenarios, APE was about 3% during the first days of forecasting and then was able to increase to 63% under the optimistic scenario, up to 200% under the pessimistic scenario and up to 100% under the most probable scenario.

Under the optimistic scenario, according to MAPE, forecasting with an acceptable accuracy of <50% is possible over the entire 100-day segment. However, in other scenarios, this opportunity is reduced to 75 and 50 days (Table 2b).

Within the third testing time frame (from 500 to 600 days), the prediction results were worse than in previous time frames (Figure 5e,f). Forecasting time with an adjusted error was shorter. In the optimistic scenario, forecasting time was 75 days; in the pessimistic scenario, it was 25 days; and in the most probable scenario, it was 50 days (Table 2c).

## 4. Discussion

### 4.1. The Findings and Their Implications

In our opinion, the problem of long-term forecasting of COVID-19 is associated with the absence of any regularities (periodicities) in its spreading. Knowledge of its dynamics in the past provides little information about its behavior in the future. For example, in [6], an analysis of new cases in three countries from 1 May 2020 to 30 November 2021 demonstrated that, in the USA, there were three peaks of daily new cases, which lasted for a long time; in India there was only one peak in that period; at the same time, the overall trend in new cases in Brazil was relatively stable. A similar aperiodic dynamic of incidence was also found in Kazakhstan. The reasons for this have been discussed above. Such processes with multiple waves of infection could never be explained using the SIR model with constant parameters. In this regard, we created an infection spread model with which we are able to create transmission rates as realizations of Gaussian random walks with unknown variances that must be learned from data.

Is it possible to use such a model for the prediction of total cases? We tested three different time frames and evaluated the optimistic, pessimistic and most likely scenarios. In all time frames, the actual numbers of total cases fell outside the predicted interval. In this case, the actual data were closest to the minimum possible forecast values, i.e., to the optimistic scenario.

In the first testing time frame, the optimistic scenario showed a high degree of forecasting accuracy, about 90%, for all 100 days. However, the prediction results were worse for the second testing time frame, which included one wave of small amplitude and an ascending section of the most “powerful” wave. We obtained an even greater discrepancy between the actual and forecast data when testing the third time frame. It was the period when the maximum peak of new cases of the disease and the slow decline in their number were registered. Therefore, our first conclusion is that the forecasting result significantly depends on the degree of incidence fluctuation in the previous period—the greater the variability of the transmission rate in the training time frame, the greater the discrepancy between the actual and model data in the testing time frame. 

Unlike previous works, we had at our disposal information on the dynamics of morbidity covering almost two years. This sizable training database allowed us to create a model with the following benefits:–Estimation of model parameters improved due to the increase in sample size.–Three scenarios of the development of the situation were proposed—optimistic, pessimistic and most probable—which is especially important for regulators when developing various response measures.–The duration of forecasting with acceptable accuracy was 100 days, which significantly exceeds some of the literature data [61]. When compared with the information from Table 3, it is obvious that the results obtained by us in terms of the duration and accuracy of the forecast surpass those of the indicated models. The already-mentioned review [9] demonstrates the high prediction accuracy of machine-learning models. However, the duration of forecasting in these works was no more than 14 days. Under the optimistic scenario, the quality of long-term forecasting of our model can be assessed as high and good.

### 4.2. Limitations

In all the tested time frames, the expected total cases were higher than the observed ones. Apparently, this is the result of the restrictive measures that were established, which were either strengthened or eased. We were not able to assess the degree of compliance with these measures by members of the society and the degree of their impact on the pandemic. Another fundamental limitation is the compliance of officially published data on the incidence of the actual situation, given the ability of the healthcare system to test the population and the quality of test systems.

The obtained discrepancies between actual and model data may be the result of a shortcoming of the proposed model itself, which is based on the use of parameter increments that had a zero-mean value and were independent of time. That is, the model does not make a priori assumptions regarding how the key parameters will change from one day to another, since making such assumptions under the influence of various social factors is a non-trivial task, if not impossible.

Table 4 presents the statistical characteristics of the transmission rate, calculated from real data in the tested periods. Mean values and standard deviations differ from the a priori assumptions used in the simulation. Previously, in Figure 4, we demonstrated that the transmission rate distribution followed the normal distribution in the training period. However, the distribution of the real transmission rate did not correspond to the normal distribution in all three testing time frames (Figure 6).

Even in the case of the stochastic nature of the transmission of the disease from one person to another, in order to make a correct prediction, we need an educated guess concerning the dynamics of *β(t)* in the future. In our case, the assumption of Gaussian random walks with estimated parameters did not lead to a reliable prediction. Differences between a priori assumptions and a posteriori reality are important causes of inaccuracies. Interesting results were obtained in [6], where white noise was used as a source of uncertainty. The authors believe that the model, on the other hand, systematically underestimated the severity of the Omicron variant of COVID-19 because of unrealistic noise-induced transitions. 

The inaccuracy of long-term forecasting can also be explained by the limitations that were imposed on the model. For example, in reality, symptomatic persons may not be isolated immediately but may continue to be a source of infection. Another aspect is the incubation period, which we adopted as 6 days. However, different authors have suggested incubation periods of COVID-19 from 3 to 11 days [70]. According to recent data, the mean incubation period of COVID-19 was 5.00 days for cases caused by the Alpha variant, 4.50 days for the Beta variant, 4.41 days for the Delta variant and 3.42 days for the Omicron variant [58]. 

Another factor that we did not consider in our model is the level of vaccination of the population. The vaccination process in Kazakhstan was irregular, and the publication of data on this process was also irregular (sometimes once a week, sometimes once every two weeks). Moreover, although it is known that the vaccinated population in Kazakhstan is 56.54%, we did not have data on the number of persons vaccinated day by day. Therefore, we assumed that it is one of the factors of stochasticity and is reflected in the irregularity of the transmission rate. Moreover, the model does not consider the possibility of recovered persons becoming reinfected. Three parameters—the incubation period, vaccination rate and the reinfection rate—can be introduced into the model as stochastic elements. All this justifies the need for the further improvement of the model of the spread of this infection.

## 5. Conclusions

Thus, the stochastic model we propose provides satisfactory results for the prediction of COVID-19 from 25 to 100 days. The information we currently have about this virus does not allow us to make predictions with high accuracy in the medium and long term. We have yet to accumulate sufficient knowledge of the disease to make any educated guesses about the patterns of its spread.

## Figures and Tables

**Figure 1 healthcare-11-00752-f001:**
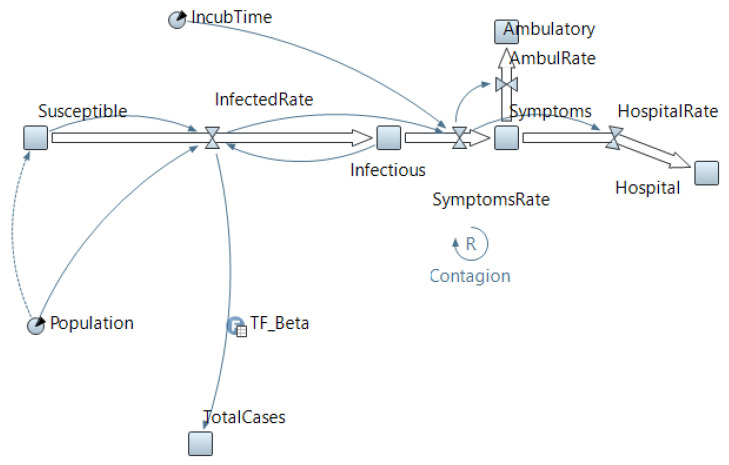
Virus spread SD model.

**Figure 2 healthcare-11-00752-f002:**
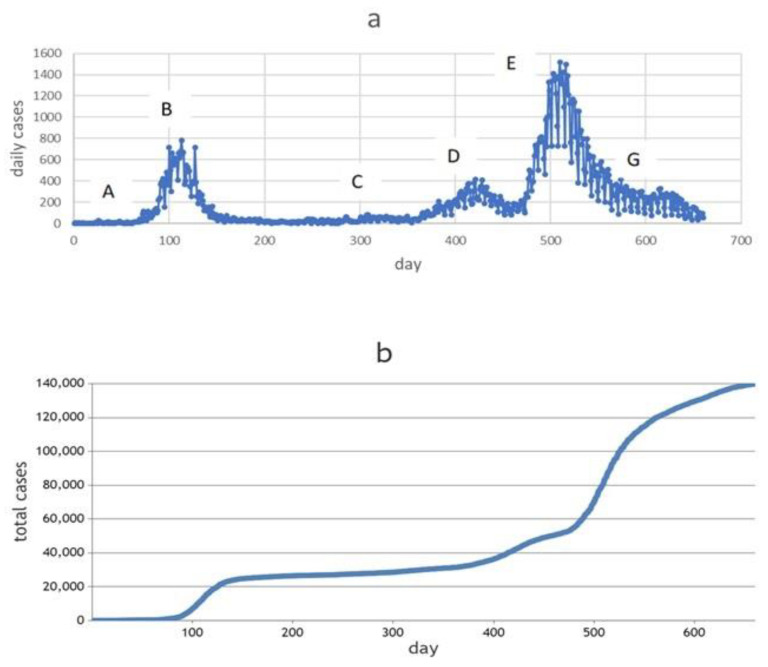
Daily cases (**a**) and total cases (**b**) from 10 March 2020 to 31 December 2021.

**Figure 3 healthcare-11-00752-f003:**
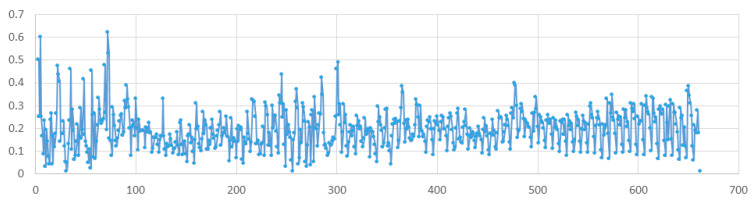
Daily calculated value of transmission rate.

**Figure 4 healthcare-11-00752-f004:**
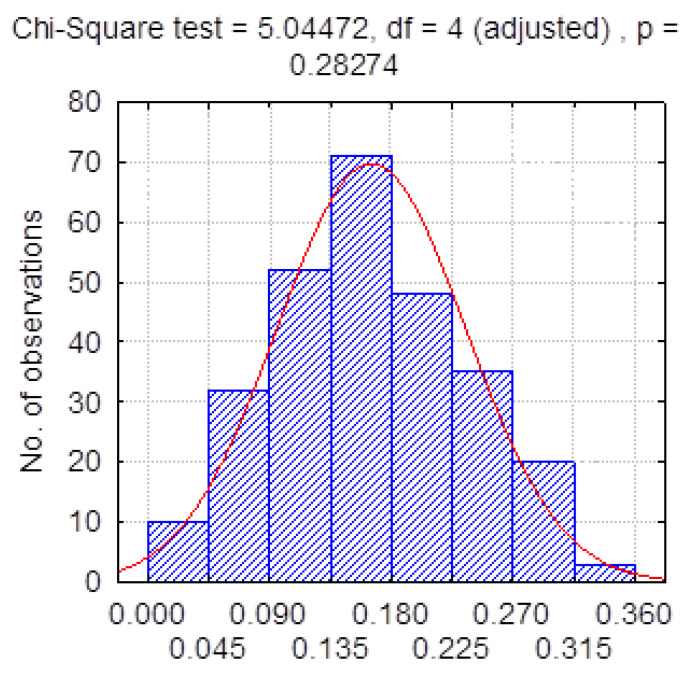
Transmission rate distribution.

**Figure 5 healthcare-11-00752-f005:**
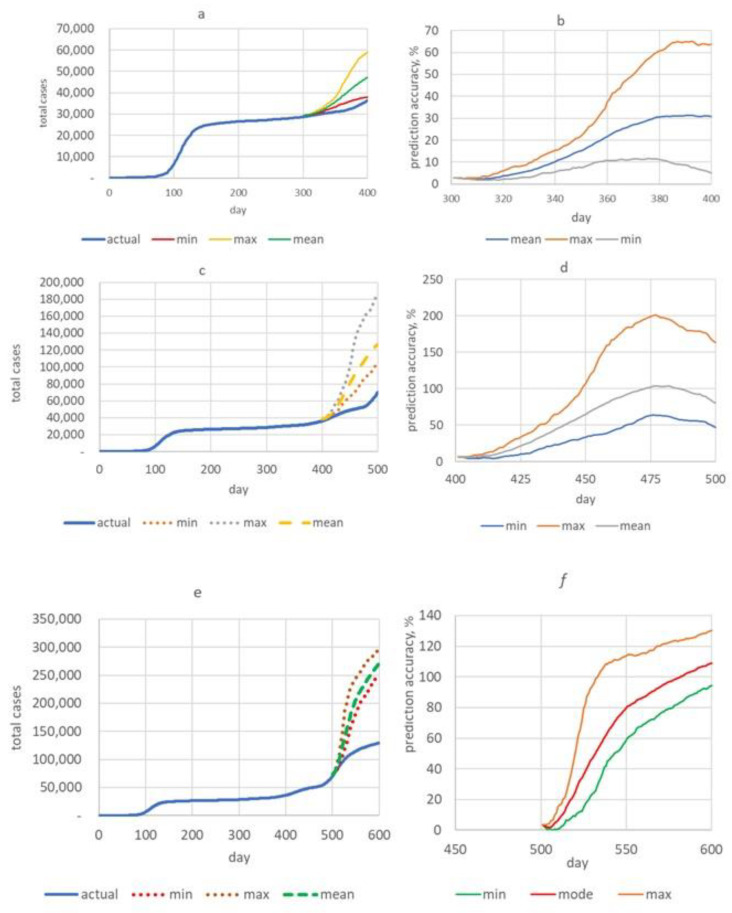
Forecasting total cases (left) and evaluation of forecasting accuracy APE (right). (**a**,**b**) First testing time frame; (**c**,**d**) second testing time frame; (**e**,**f**) third testing time frame. Green line—optimistic scenario; orange line—pessimistic scenario and red line—most likely scenario.

**Figure 6 healthcare-11-00752-f006:**
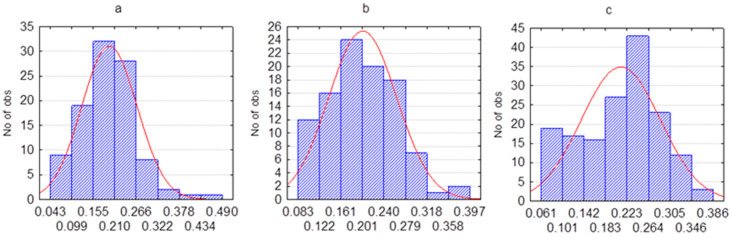
Transmission rate normality checking on the various testing time frames. (**a**) First testing time frame, 300–400 days (Shapiro–Wilk test, *p* = 0.00063). (**b**) Second testing time frame, 400–500 days (Shapiro–Wilk test, *p* = 0.049). (**c**) Third testing time frame, 500–600 days (Shapiro–Wilk test, *p* = 0.00382).

**Table 1 healthcare-11-00752-t001:** Calculated values for transmission rate at the different training time frames.

Training Time Frame, Day	1–300	1–400	1–500
*β* (mean, SD)	0.17 (0.075)	0.177 (0.074)	0.182 (0.073)

**Table 2 healthcare-11-00752-t002:** Forecasting accuracy (MAPE%) depending on the duration of the time frame.

MAPE (%)				
Duration of Forecasting	25 Days	50 Days	75 Days	100 Days
(a) First testing time frame				
Optimistic scenario	2.235	3.712	5.986	6.697
Pessimistic scenario	4.481	9.400	19.781	30.569
Most likely scenario	3.004	6.320	11.897	16.629
(b) Second testing time frame				
Optimistic scenario	6.209	14.119	25.059	33.154
Pessimistic scenario	16.532	40.448	83.038	108.475
Most likely scenario	10.405	26.836	46.881	59.193
(c) Third testing time frame				
Optimistic scenario	5.648	22.237	38.360	50.641
Pessimistic scenario	29.947	65.941	83.094	93.817
Most likely scenario	14.143	37.537	54.659	66.829

**Table 3 healthcare-11-00752-t003:** Some previously reported forecast accuracy metrics.

S. No.	References	Forecasting Techniques	Duration of Forecasting	MAPE Score
1	Gupta, R.; et al., 2020 [62]	SEIR and regression models	20 days	SEIR: 25.533, Regression: 21.889
2	Khan, F.M.; Gupta, R., 2020 [63]	ARIMA and NAR	50 days	ARIMA: 362.1761
3	Sujath, R.; et al., 2020 [64]	LR, MLP and VAR	69 days	LR: 1745454.432, MLP: 80.057, VAR: 43289.290
4	Tiwari, S.; et al., 2020 [65]	Time series forecasting using Weka	24 days	55.489
5	Tomar, A.; Gupta, N., 2020 [66]	LSTM	25 days	63.357
6	Salgotra, R.; et al., 2020 [67]	Genetic programming-based model (GP) (GEP model)	10 days	7.827
7	Sunori, S.K.; et al., 2021 [68]	Exponential growth model and ML-based LR model	33 days	LR: 662.441, Exponential: 2096.000
8	Mr. Sudip Ghosh, 2020 [69]	Least square fit- ted model	35 days	39.816

**Table 4 healthcare-11-00752-t004:** Transmission rate statistical characteristics for actual and modeled data.

	Time Frame, Day	300–400	400–500	500–600
Actual data	*β* (mean, SD)	0.195 (0.071)	0.200 (0.061)	0.204 (0.065)
Model data	*β* (mean, SD)	0.17 (0.075)	0.177 (0.074)	0.182 (0.073)

## Data Availability

The data presented in this study are available upon request from the respective author. Data accessibility requires permission from the Ministry of Health of the Republic of Kazakhstan.

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
