# Peer review of "Is It Possible to Predict COVID-19? Stochastic System Dynamic Model of Infection Spread in Kazakhstan"

_healthcare, 2023, doi:10.3390/healthcare11050752_

Round 1
Reviewer 1 Report
Please see the attached file.

Reviewer 2 Report
Reviewer Report for “Is It Possible to Predict COVID -19? Stochastic System Dynamic Model of Infection Spread in Kazakhstan” submitted to Healthcare.
The authors of the manuscript proposes a stochastic system dynamic model to forecast COVID-19 spread. The SIR model is applied to model spread in Kazakhstan. The empirical evidences show that the proposed model has a very good forecasting power. It is an up to date study and can be published after some minors. Please consider the followings:
Also, in the literature review part could you please compare the study with the existing one in the literature. Also, please consider the work
https://doi.org/10.1080/02664763.2022.2028744
- It is an up to date study and it enlighten the dynamics of Covid spread in stochastic senses.
What does it add to the subject area compared with other published material? This comperision should be done by the authors.
Please compare the existing literature with the proposed methodology. What do you offer new in this study.
Could you please emphasizes the novelty of the study in the introduction? What is different than the SIR model please clarify the subject.
Representation of the tables and figures are adequate In the conclusion part please discuss the limitation of the study. Also please provide some insights for the decision makers.
Reviewer 3 Report
The article is devoted to developing a stochastic model based on the system dynamics approach for predicting COVID-19. The study's relevance is justified by the fact that with the onset of the COVID-19 pandemic, scientists began to actively use models to determine the epidemiological characteristics of the pathogen. The rate of transmission, recovery, and loss of immunity to the Covid-19 virus change over time and depend on many factors such as the seasonality of pneumonia, mobility, frequency of testing, use of masks depending on the weather, social behavior, stress, and public health measures, etc. This study aimed to predict COVID-19 using a stochastic model based on a system dynamics approach. The authors propose a modified SIR model in the AnyLogic program. The key stochastic component of the model is the rate of transmission, which the authors consider as an implementation of Gaussian random walks with unknown variance obtained from real data. The actual data of the total number of cases was outside the predicted minimum-maximum interval. The minimum predicted values of the total number of cases are closest to the real data. The stochastic model proposed in this study gives satisfactory results for short-term, up to 20 days, forecasting of COVID-19. Forecasting for more prolonged periods has an error of more than 10%. In our opinion, the problem of long-term forecasting of COVID-19 is associated with the absence of any reasonable assumptions about the dynamics of β(t) in the future. The proposed model needs to be improved by removing limitations and including more stochastic parameters.
Despite the satisfactory quality of the article, some shortcomings need to be corrected.
- In the introduction section, the authors review low-precision models. This looks like an unreliable coverage of the real picture because a large number of models have an accuracy of more than 95%. Among such models are those based on statistical machine learning (for example, Random Forest, K-Nearest Neighbors, Gradient Boosting) and deep learning. Such publications should be included in the review in descriptions of a more realistic picture.
- It needs to be clarified which problem solves authors with the proposed model. It is recommended to define the drawbacks of the described models and connect these drawbacks with the proposed solution.
- The structure of the proposed model should be justified. It should be described why the authors include such compartments.
- It is unclear why authors provide differential equations only for the Susceptible state, not for all systems.
- Limitations that authors applied to the model should be justified. It is unclear why “the recovered population does not become susceptible again”, because many cases of re-infection are known. The statement “the incubation period for all infected is 6 days” does not correspond with the last research.
- It needs to be clarified why authors do not consider vaccination. It is the most valuable control measure for COVID-19.
- The authors calculate a retrospective forecast but do not estimate its accuracy. It should be evaluated using different metrics.
- The authors implemented a simple compartment model with AnyLogic software. The scientific novelty of the research should be highlighted.
- The practical aspects of introducing the proposed model to healthcare facilities should be discussed.
- The English should be proofread.
- The formulas are parts of the sentences. The punctuation should be corrected.
In summarizing my comments, I recommend that the manuscript is accepted after major revision.
Round 2
Reviewer 1 Report
No comments.
Reviewer 2 Report
The manuscript can be published in this form.
Reviewer 3 Report
Thanks to the authors for considering some reviewers' comments and recommendations. Hovewer some drawbacks that still needed to be eliminated:
- The authors provided differential equations only for S, I, and R states (page 5). The model should contain differential equations for all states of the proposed model.
- Authors say that data on the incubation period varies widely: from 1.8–9.0 days according to a review [1], from 1 to 14 days according to [2], the average value was 5.8 days (95% CI from 5.0 to 6,7) in [3]. However, references were published in 2020, and the incubation period of COVID-19 has changed due to virus mutations and more investigations provided. It is recommended to include the justification of the incubation period not only in the answer but also in the text of the article.
- According to WHO data, the vaccinated population in Kazakhstan is 56.54%, significantly affecting the dynamics of the epidemic process. Therefore, authors should either include vaccination in the model or justify why vaccination is not included and formulate the model's limitations caused by this.
- Expanding the current research analysis, including papers on ML applications for COVID-19 forecasting, is still recommended. E.g. doi: 10.1016/j.chaos.2021.111779, doi: 10.1136/bmjopen-2021-056685, doi: 10.3390/computation10060086, doi: 10.3389/fpubh.2021.729795
Author Response
Thank you for your review of our paper. We have answered each of your points below

Round 3
Reviewer 3 Report
Thanks for the authors for considering reviewer's comments and recommendations. In my opinion, the paper can be accepted.